# A Narrative Review of the History of Skin Grafting in Burn Care

**DOI:** 10.3390/medicina57040380

**Published:** 2021-04-15

**Authors:** Deepak K. Ozhathil, Michael W. Tay, Steven E. Wolf, Ludwik K. Branski

**Affiliations:** Department of Surgery, University of Texas Medical Branch at Galveston, Galveston, TX 77550, USA; mitay@utmb.edu (M.W.T.); swolf@utmb.edu (S.E.W.); lubransk@utmb.edu (L.K.B.)

**Keywords:** skin graft, history, autograft, burn, dermatome, mesh, split-thickness, xenograft, CEA, CSS, Spray-on-Skin, ReCell

## Abstract

Thermal injuries have been a phenomenon intertwined with the human condition since the dawn of our species. Autologous skin translocation, also known as skin grafting, has played an important role in burn wound management and has a rich history of its own. In fact, some of the oldest known medical texts describe ancient methods of skin translocation. In this article, we examine how skin grafting has evolved from its origins of necessity in the ancient world to the well-calibrated tool utilized in modern medicine. The popularity of skin grafting has ebbed and flowed multiple times throughout history, often suppressed for cultural, religious, pseudo-scientific, or anecdotal reasons. It was not until the 1800s, that skin grafting was widely accepted as a safe and effective treatment for wound management, and shortly thereafter for burn injuries. In the nineteenth and twentieth centuries skin grafting advanced considerably, accelerated by exponential medical progress and the occurrence of man-made disasters and global warfare. The introduction of surgical instruments specifically designed for skin grafting gave surgeons more control over the depth and consistency of harvested tissues, vastly improving outcomes. The invention of powered surgical instruments, such as the electric dermatome, reduced technical barriers for many surgeons, allowing the practice of skin grafting to be extended ubiquitously from a small group of technically gifted reconstructive surgeons to nearly all interested sub-specialists. The subsequent development of biologic and synthetic skin substitutes have been spurred onward by the clinical challenges unique to burn care: recurrent graft failure, microbial wound colonization, and limited donor site availability. These improvements have laid the framework for more advanced forms of tissue engineering including micrografts, cultured skin grafts, aerosolized skin cell application, and stem-cell impregnated dermal matrices. In this article, we will explore the convoluted journey that modern skin grafting has taken and potential future directions the procedure may yet go.

## 1. Introduction

“(A Spaniard) upon a time walked in the field, and fell at words with a soldier, and began to draw (his sword); the soldier seeing that, struck him with the left hand, and cut off his nose, and there it fell down in the sand. I then happened to stand by, and took it up, and pissed thereon to wash away the sand, and dressed it with our balsama artificiato, and bound it up, and so left it to remain 8 or 10 days, thinking that it would have come to matter; nevertheless when I did unbind it I found it fast conglutinated, and then I dressed it only once more, and he was perfectly whole.”—Leonardo Fioravanti [1].

The above is an excerpt from the sixteenth century works of Leonardo Fioravanti (1517–1588), a charismatic Italian surgeon, who was controversial in his time for his vocal rejection of Galenic doctrine and credited with performing the first splenectomy on Italian soil [1]. Fioravanti was also the first in the western world to document the successful reattachment of a severed body part, in this case a nose. The excerpt describes a 29-year-old gentleman named Signor Andreas Gutiero living in Africa, who, much to his own misfortune, chose to engage in a heated argument with a soldier stationed there. As described above, this interaction did not go well and left Gutiero breathless and detached from his nose. Leonardo Fioravanti happened upon this encounter and was able to achieve an outcome that would be considered remarkable for any time period, and certainly in his. Though this anecdotal story is not a true description of a skin grafting per se, the practice of tissue restitution lays the foundation for the development of modern skin grafting. Following Fioravanti’s report, numerous similar descriptions of nasal restitution subsequently appear throughout the literature as criminal punishment with nasal disfigurement was a common practice in those days. Arguably, auto-transplantation of tissue to its own donor site is the precursor to cutaneous autografting.

Burn injuries and their treatments are intertwined with human history dating back to the origins of mankind’s relationship with fire. In fact, the rich history of burn wound treatments predates civilization. Archeologists have found cave paintings depicting Neanderthal-man having treated burn wounds with plant-based extracts [2,3]. Numerous concoctions over the ages have been utilized to treat burn wounds. In the ancient world, the Ebers Papyrus (1500 BC) describes Egyptian physicians making salves derived from animal, plant, and mineral byproducts and combining their application with religious ceremonies to the Goddess Isis. Burn wounds were then dressed with bandages moistened with the milk derived from the mothers of male infants [4]. The Romans had a pharmacopeia of products to treat burn wounds, ranging from mixtures of honey and bran to cork and ashes [3]. For millennia, physicians have attempted to treat burn wounds with all manner of products and combinations therein, but were met with middling success due to the lack of scientific understanding about burn wound pathophysiology. Independent of burn injuries, the history of skin grafting followed a similar trajectory with limited success for several centuries due to a combination of inefficient tissue collection methods, inappropriately thick grafts, and a lack of understanding of the physiology behind skin grafting. It would not be until the nineteenth century on the wave of numerous medical advances in burn care that the Swiss surgeon Jacques-Louis Reverdin (1842–1929) and the English surgeon George Pollock (1897–1917) would first apply skin grafting techniques to the treatment of burn wounds [5]. 

Modern burn care is the result of numerous advances in wound care, understanding of burn sepsis pathophysiology, operative technology, and surgical technique. The single most significant advancement credited for heralding the modern age of burn care is the utilization of skin grafting after early wound excision first introduced in the 1940s [6]. Today, autografting of full thickness burn wounds is the standard of care, having a direct effect on time-till-wound-closure, and a substantial impact on morbidity and mortality for burn victims. In this historical review, we will trace the evolution of surgical techniques, the development of operative instruments and the advancement in physiologic knowledge about skin grafting through the ages. We hope to integrate a common thread of the lessons hard-learned by numerous exceptional surgeons in our timeline to best appreciate how the state of modern burn care came to be. Furthermore, the author hopes to explore techniques and cutting-edge technologies that are anticipated to play a significant role in the burn care of tomorrow.

## 2. Skin Grafts in Antiquity: 3000 BC–476 AD 

Facial mutilation was a common punishment in the ancient world and practiced in much of Asia and Europe. It was often performed by cutting off the nose or ear of victims as a punishment for crimes committed, but also served as a warning to other would-be wrong-doers. The silver lining of centuries of this painful and humiliating practice was that it inspired the development of skin grafting [7]. One of the oldest descriptions of nasal mutilation comes from an ancient Indian Sanskrit epic from 1500 BC, the Ramayana, in which Lady Surpunakha (Meenakshi), angry after being scorned by Prince Rama, attacks his wife Princess Sita. As punishment, her nose is amputated by Rama’s brother, Prince Lakshmana. The cultural significance of this act is that the nose is synonymous with respect. As a consequence, King Ravana orders her nose be reconstructed [7]. Although this legend leaves many details to the imagination, it highlights how commonplace such practices were. For example, in 1769 the Ghoorka King of India captured the city of Kirtipoor in modern day Nepal. He ordered the nasal mutilation of all 865 male inhabitants and changed the name to Naskatapoor, which translates to “city without noses” [8]. The cultural influence of more than three millennia of this practice is undeniable, exemplified by idiomatic expressions such “loosing face” which alludes to a loss of dignity and wonton embarrassment. In Urdu and Punjabi there is a colloquial expression “mera noc kart gaya” meaning “you have hurt my feelings”, but literally translated as “you have cut off my nose” [7]. It is therefore no surprise that nasal reconstruction is the oldest form of facial surgery.

The first operative description of tissue translocation was performed by the Indian surgeon Sushruta (approximately 750–800 BC), considered by many historians to be the “Father of Indian Surgery”. Indian surgeons (referred to as Hindoo surgeons in early texts) underwent an extensive period of training on anatomy and hand dexterity. He described the instruments and surgical techniques he used in Sanskritt hymns called Vedas. The best known of these is the Sushruta Samitá and serves both as an educational and religious text [5]. Sushruta outlined the progenitor for the modern pedicle flap by rotating and advancing tissues from the cheek [9]. This method, now called the “Indian cheek flap”, is the oldest documented method of skin translocation and is sometimes referred to as the “Indian Method” of nasal restitution. Sushruta also documented more than 15 methods to repair mutilated ears and lips. Our modern understanding of skin grafting originates directly from the Indian Method as it would be distributed and discovered several times over the years. The Indian Method by way of migrating surgeons would make it is way to Egypt, Greece, Arabia (the Middle East), and ultimately Italy over the centuries. Knowledge of the procedure would be discovered and credit misappropriated several times. The “French Method” for example, is considered to be a recapitulation of Sushruta’s techniques several centuries later. The Indian cheek flap, in our modern nomenclature, would be considered an advancement of sliding flap [10].

In the time of Buddha (562–472 BC), progress in Indian surgery came to a standstill because of social and religious pressures. Although great respect to the field of medicine was afforded by the Buddhists, animal experimentation and direct contact with bodily fluids and diseased tissues was considered to be spiritually defiling [10]. Therefore, surgical responsibilities fell by default to the second lowest cast in the Hindu system, the Shudras. Called by various names (Koomás), and described later by Europeans based on their profession (potters, bricklayers, and tile-makers), the Shudras were considered unclean and therefore not at risk of being further defiled by the handling of blood and pus, necessary in the practice of surgery. Being generally uneducated compared to the surgeons of Sushruta’s day, much fundamental surgical wisdom would be lost for an age. 

The origin of the techniques practiced by the Shudra is controversial and discrepant by several millennia, depending on the source, due to the entanglement of surgical folklore and oral traditions—3000 BC, 600 BC, 1000 AD, and 1440 AD have all been reported. [5,11,12]. Their techniques, now more commonly referred to as the “Ancient Indian Method”, actually referred to two different procedures. The first procedure would be described as a median forehead pedicle flap. The patients were operated upon awake and in an upright position to minimize blood-loss and protect their airway. In addition, a handkerchief around the neck would reportedly be used to induce transient venous congestion. This procedure was first described in 1794 by the British Army surgeon Cully Lyon Lucas, who was stationed in Madras (Chennai, India). At the time, Tipú Sultán, the ruler of Mysore (Karnataka, India) waged a guerilla war against the British Crown by placing a bounty on the noses of any Indian’s who helped transport grain for the British. Lucas described in detail a median forehead flap for nasal restitution performed on one such victim named Cowasjee, who had his nose and one hand mutilated. He would publish his observation in the Gentleman’s Magazine (London, UK) [13]. The reconstructed “Hindoostan noses” were subsequently observed by others, who commented on their likeness to native nasal tissue and sturdiness to tolerate sneezing and nose-blowing [5]. 

The second procedure would be considered a free full-thickness skin graft. In this technique, a wooden sandal would be used to percuss the gluteal region until it had been contused and inflamed sufficiently to generate local edema. Then, the skin and underlying subcutaneous fat would be harvested and applied immediately to the nasal wound, along with a proprietary surgical cement that was described to have healing properties [12]. These procedures, as well as the equipment and products used at the time (topical hemostasis, cotton sutures, intranasal splints, various leaves, and ant-heads as skin staples), were closely guarded secrets passed down from father to son within families [5].

Stories of these methods slowly made their way to ancient Egypt and Rome, carried by “students and itinerant surgeons” [12]. In 1600 BC, the Edwin Smith Papyrus is a record of a number of ancient Egyptian treatments for those who have suffered mutilations to the face, though none would be considered reconstructive. In the first century, the Roman encyclopedist Aurelius Cornelius Celsus (25 BC–50 AD) wrote De Medicina, which depicted a multitude of skin flaps used to repair ears, noses, and lips [14,15]. Celsus even described how to reconstruct foreskin to enable circumcised Jewish men to be more accepted by the Romans [5]. In the second century, the famous Roman surgeon Claudius Galen (129–210 AD) described a number of procedures to reconstruct facial injuries with local tissue advancement flaps. The works and theories of Galen would form the Galenic doctrine, which would make up the mainstay of medical education and practice for the next thousand years [16]. 

## 3. Skin Grafts in the Middle Ages and Renaissance: 476–1789 AD

The majority of early medical knowledge in Europe during the Middle Ages rested with the Catholic Church and was preserved in monastic texts. In 1215 AD, however, Pope Innocent III banned any priest, deacon, or sub-deacon from performing any surgical procedure due to the religious misgivings that bloodshed was “incompatible with the divine mission” [10]. Following the Pope’s decree, and quite similar to the impediments that befell surgical practice in ancient India, academic progress in Europe came to a stand-still and ultimately much surgical wisdom was lost for several centuries [10].

In the eighth century, the Sushruta Samhitá was translated into Arabic and along the silk road made its way to Italy. The first European surgeon to practice the cheek-flap technique was Gustavo (Branca) de’Branca in Catania (Sicily) in the fifteenth century. His son, Antonio Branca further developed the procedure into a six-stage rhinoplasty that utilized a myocutaneous flap from the arm. This method remained a closely held secret within the Branca family for nearly a century. Though first documented in 1460 by Heinrich von Pfolspeundt, a knight of the Teutonic Order, it would not be until 1493 that a detailed description of the procedure would be make available to the public by Alessandro Benedetti (1450–1512) a professor at Padua University (Italy). More than a hundred years later, the famous Italian surgeon Gaspare Tagliacozzi (1545–1599), while practicing at the Hospital of Death would cite Benedetti and build upon the Branca technique in his own work De Curtorum Chirurgia per Insitionem (published in 1597). For his contributions, Tagliacozzi is considered a pioneer in reconstructive and plastic surgery and credited for the “(Ancient) Italian Method” of tissue translocation [17].

Not surprising for the time, Tagliacozzi incurred the antagonism of the Catholic Church, which viewed reconstructive surgery as a form of sacrilege that meddled with God’s creations [2]. His writings were, therefore, declared heretical and burned by decree. The Church even went so far as to exhume Tagliacozzi’s body in order to rebury in unconsecrated ground [7]. As a few copies of his writings survived the purge, a number of Tagliacozzi’s followers attempted to reproduce his techniques. Unfortunately, they could not replicate his results. After numerous failed attempts and the slow passage of time, Tagliacozzi’s reputation would slowly slide into mockery and denigration. For nearly 200 years, little progress would be achieved. In fact, by the end of the eighteenth century the Paris Academy of Surgery would declare Tagliacozzi’s (Italian) Method impossible. Once again, we see history repeat itself as short-lived advances in skin translocation are prematurely abandoned due to social and religious pressures of the age. 

In the sixteenth century the French Barber–Surgeon Ambroise Paré (1510–1590 AD), who served in the royal courts of Kings Henry II, Francis II, Charles IX, and Henry III, wrote voluminously on the importance of procedural interventions to reduce pain and suffering. His efforts are thought to have laid the foundation for empiricism and the modern concept of evidence-based practice. In addition, Paré helped prompt the transition of Barber–Surgeons to their contemporary status as physicians [18]. Barber–Surgeons were entrusted with performing operations under the supervision of physicians due to their familiarity with handling blades. Galenic doctrine had been the mainstay of western medical education for more than a thousand years and it emphasized theory over empirical knowledge [18]. For example, the recommended treatment for burn wounds from gunpowder was to ‘detoxify’ the patient by cauterizing the wounds with boiling oils. The method was so common that it was referred to in Shakespeare’s *King John*, where Cardinal Pandulph says to King Philip:

“And falsehood falsehood cures; as fire cools fire within the scorched veins of one new-burned.”King John (Act 3, Scene 1).

Paré was trained at the renowned Hôtel Dieu and was appointed as a surgeon in the military directly out of his training. Perhaps fortunately for Paré, the beginning half of the sixteenth century was a time of great turmoil in Europe. The ambitious French king, François I, seeking to increase his domains and influence waged multiple wars against the industrious Emperor of Germany, Charles V. One such battle took place in 1535 when Anne de Montmorenci, commanding the armies of France, pursued the retreating imperial army from Provence to the Pas de Suze (the Suza Pass). It was on this campaign through the Alps that Paré first gained experience caring for burn injuries. At the time, Paré served under the inferior title of “surgeon” as he had not yet been confirmed into the ranks of Barber–Surgeons, having been sent to the Italian front prior to completing his certifying examinations. Ironically, however, it was as a surgeon that Paré made many of the observations for which he is credited for today [19]. 

During the battle of Pas de Suze, Paré ran out of boiling oil, the recommended treatment for burn wounds. He, therefore, improvised and created a soothing balm made from egg yolks, rose oil, and turpentine. The following morning, he found that the patients treated with the balm were resting comfortably, while those treated in the traditional manner remained ill and febrile [20]. This simple observation would change the fundamental practice of medicine and the treatment of burn injuries from theory-based to evidence-based practice. Paré, upon seeing the dramatic difference in his patient groups, broke from his Galenic teachings. He would go on to characterize various degrees of burn wounds and contractures, and even pioneer the practice of burn wound excision [21]. Although Paré did write about skin grafting, he viewed its benefits with considerable skepticism [10]. This is not surprising for two important reasons. First, knowledge of burn wound pathophysiology remained fundamental and severely injured patients failed to receive appropriate resuscitation that would be considered acceptable by modern standards. Second, access to surgical interventions was limited, especially for those who would benefit from it the most. As general anesthesia and the technique of transfusing blood had not yet been invented, surgical excision of large burns was not considered possible. Therefore, Paré’s perspective was influenced by a selection bias — that is to say patients who survived long enough to be considered for surgery would likely continue to survive independent of the treatment they then received. Nonetheless, his efforts to promote evidence-based wound care would play an incredibly important role in the development of modern skin grafting. 

## 4. Skin Grafts in the Early Modern Era—Age of Revolution: 1789–1849 AD

The nineteenth century was unique in that it marked the dawn of the Contemporary Age. Important notable advancements in medicine during this time included the discovery of anesthesia and the dawn of microbiology. The introduction of general anesthesia is credited to the Boston dentist William Thomas Green Morton (1819–1868), who on 16th October 1846 performed his famous demonstration with diethyl ether at Massachusetts General Hospital on Mr. Edward Gilbert Abbott during the excision of a cervical mass. The introduction of anesthesia made it possible to perform more invasive procedures including larger surface area skin grafting, with improved safety and tolerance to the patient. 

Another momentous achievement came through Louis Pasteur and Joseph Lister’s work on microbiology. Their research on antiseptic techniques revolutionized anti-microbial wound care and improved post-operative outcomes [10]. Riding on the coattails of scientific breakthroughs in multiple fields including microbiology, anatomy and physiology, medicine itself was dramatically changing in response to evidenced-based practices. Both the Indian and Italian methods would see their revivals during this period. The English surgeon Joseph Carpue (1746–1840) recapitulated the Ancient Indian Method (median forehead flap), as originally documented by Lucas in 1816, effectively reviving it in Europe [5]. Similarly, the German surgeon Karl Ferdinand von Gräfe (1787–1840) would successfully revive the Italian Method in 1817 [2]. 

The Indian and the Italian Methods of nasal reconstruction, though important in the origin of the (free) skin graft, were nonetheless technically not skin grafts themselves, but pedicle flaps. The transition from the pedicle flap to the skin graft is not, however, as linear a narrative as one would imagine. Although potentially first performed by the Shudras of India (Ancient Indian Method), credit or even proof of their achievements cannot be well validated as there is no specific documentation of their surgical technique or outcomes from prior to the nineteenth century. In 1941, the American surgeon Sumner L. Koch (1888–1976) wrote [22]:

Although it has been said that in India the restoration of the nose with the free grafts of skin and subcutaneous tissue was successfully accomplished. I have been unable to find any definite record of it and if it was actually carried out it remains an achievement that the surgeon of today has not been able to equal. Sumner L. Koch [22].

It would not be until the latter half of the nineteenth century that surgeons like Jacques-Louis Reverdin, George Pollock, and John Reissberg Wolfe would successfully perform and proselytization the proper technique for (free) skin grafting. To this affect, in 1874 Wolfe wrote the following:

This pedicle (flap) has, in my opinion, been a source of great embarrassment to surgeons, and tended rather to retard the progress of plastic surgery. (…) I have long held it demonstrated, that in most cases the pedicle is not essential, if indeed it (does) contribute anything, to the vitality of (the) flap. John R. Wolfe [22].

Wolfe’s beliefs, however forward thinking for his day, were built upon the works of pioneers in the first half of the nineteenth century. The British surgeon Sir Astley Paston Cooper (1768–1841), for example, transferred a full-thickness skin graft from a severed thumb in order to cover the amputation stump in 1817 [23]. The German surgeon Johann Friedrich Dieffenbach (1792–1847) wrote voluminously on tissue transplant techniques to reconstruct various body parts mutilated by a number of etiologies including burns [11]. Dieffenbach is also recognized for his numerous animal experiments. In his book Surgical Observations on the Restoration of the Nose and on the Removal of Polyps and other Tumors from the Nostrils (London, 1833), Dieffenbach writes about his failed attempts to re-attach severed bodily appendages of different mammalian species: tails from cats and dogs, ears from dogs and rabbits, and even human fingers. Dieffenbach also dabbled in xenografting—transplanting pigskin to pigeons for example [24]. Despite a handful on controversial exceptions, the vast majority of Dieffenbach’s experiments were met with failure. 

The separate history of zoologic experimentation and skin xenografts is worth mentioning here as it is likely nearly as influential, if not more so, than nasal restitution in the origin or (free) skin grafts. Unfortunately, as many early zoological experiments resulted in failure, the literature is sparsely populated with confirmed experiments. In 1663, Robert Hooke (1635–1703), a British scientist for the Royal Society of London, performed a handful of successful skin transplants on chickens and dogs [25]. In the nineteenth century, a number of surgeons experimented with live xenografts (inter-species pedicle flaps) between various animals (cats, dogs, rats, rabbits, sheep, birds, and frogs) and human subjects. In these experiments, the donor animal was typically immobilized and affixed to the patient for several days in an effort to promote vascularization of the xenografted tissue [26,27]. Despite some disputed claims of success, the majority of these experiments are also believed to have failed. 

It would not be until the early nineteenth century when the Italian physician Giuseppe Baronio (1759–1811) would successfully and reproducibly perform a series of (free) skin grafts on sheep [28]. Baronio was an interesting individual with eclectic interests, underappreciated in his time, who failed to achieve promotion and died unmarried at a young age. His interest in autologous skin translocation derived directly from the nasal reconstruction techniques of the Branca family and Tagliacozzi mentioned earlier. In 1804, Baronio describes in his book, *Degli Innesti Animali* (On Grafting in Animals), the three experiments he performed at the estate farm of Count Anguissola of Albignano (Milan, Italy). In the first experiment, Baronio harvested skin from the back of a ram and immediately grafted it to a new site on the back of the same sheep. The graft was secured with adhesive dressings rather than sutures. In the second experiment, a similar procedure was performed on the same unfortunate animal, but with an 18 min delay between graft collection and placement. In the third experiment, this delay was extended to one hour. Not surprisingly, the first graft took perfectly, the second incurred some inflammation and likely superficial necrosis. The third graft failed altogether. The surviving grafts were cut into 10–12 days after transplantation and were noted to be well vascularized [29,30]. Baronio went on to perform similar experiments on 27 animals (sheep, goats, dogs, a horse, and a cow) in total. Unfortunately, his accomplishment went unnoticed, and he died in relative obscurity, but his findings mark the first scientifically documented reports of successful autologous skin grafts performed in mammals.

The German surgeon Christian Bünger (1782–1842) was the first to theorize that free tissue transfers may also be viable in humans. In 1823, Bünger performed a rhinoplasty on a 33-year-old woman named Wilhelmina in Marburg, Germany. The patient was considered to be quite beautiful at one time, but suffered from a skin condition resulting in severe disfigurement to her face. The lack of usable adjacent tissue on the face obviated the use of an advancement flap. Using a variation of the Ancient Indian Method (full-thickness skin graft), Bünger transferred tissue from her buttock to reconstruct the patient’s nasal defect. This was done with a much thinner graft free of any subcutaneous tissue and likely without the preparatory tissue percussion recommended in the original technique. This operation was applauded by many as the first verified (free) skin graft performed in Europe. However, controversy surrounded Bünger’s achievement as some stated that the surgery had only been partially successful. Furthermore, Bünger’s contemporaries would not successfully replicate his results for more than a decade [12,31].

In the United States, Jonathan Mason Warren (1811–1867) was a Harvard trained surgeon who is best remembered for being the first surgeon to administer anesthesia to a pediatric patient. In 1834, he also became the first surgeon in North America to reconstruct a nose using a median forehead pedicle flap, and in 1840 he became the first surgeon to successfully use a (free) skin graft to reconstruct a nose. Warren was a fruitful writer, who recorded his use of full-thickness skin grafts to repair eyelids and noses. In 1844, Joseph Pancoast (1805–1882) from Philadelphia, a contemporary of Warren’s, described the reconstruction of an earlobe with a (free) skin graft in his Treatise of Operative Surgery [30]. 

## 5. Skin Grafts in the Late Modern Era—Preceding World War II: 1850–1938 AD

On 8th December 1869 Swiss surgeon Jacques-Louis Reverdin (1842–1929) reported the successful use of [free] skin grafts on granulating wounds to the Société Impériale de Chirurgie in Paris, France. He described taking extremely small and very thin pieces of skin, which he called “epidermic grafts” and placing them on the granulating wound bed to act as centers of epithelization [32]. His first case involved a man who had injured his thumb. Reverdin placed two 1 mm pieces of skin onto the wound. Then, two weeks later, he observed that each graft formed a small island of epithelization and with the same amount of epidermal proliferation. Therefore, he attempted to increase the surface area treated by increasing the number of (small) grafts applied [5]. Because Reverdin’s technique involved picking up skin with forceps and excising a small piece with scissors, it became known as “pinch grafts”. Reverdin’s accomplishment were paradigm changing and gained him the reputation of Father of Skin Grafting. Nonetheless, his technique had some significant shortcomings. Due to the small size of the grafts, large wounds would suffer significant contracture. Due to the friability and the prolonged time till wound closure, it was less effective around joints. Lastly, it was not an aesthetically pleasing graft, forming bumps and pits at both the donor and graft sites [12]. Some historians also attribute credit to Reverdin for the use of skin grafting in burn wounds because he is reputed to have used skin from his own arm as allograft to treat the burn wounds on a patient’s back. Many historians, however, question the validity of this story.

Credit for treating burn wounds with skin grafts lies with George Davis Pollock (1876–1950), who presented his work titled “Cases of Skin Grafting and Skin Transplantation” before the Clinical Society of London in 1870. In his paper, Pollock described a series of 16 cases where he applied thin skin grafts to open wounds, of which 8 were successful. Pollock credited Reverdin in his work, but his very first case, an eight-year-old girl named Anne, is the first documented successful report of the use of skin grafts for the treatment of burn wounds. Anne suffered significant burn injuries to her lower extremities after her dress caught on fire. After two years of insufficient wound care, she presented to Pollock with a large persistent ulcerations on the right thigh. Pollock applied two small pieces of skin from the patient’s abdomen to the ulcer and noticed six weeks later that the ulcer had healed considerably with the small skin grafts acting as a node for secondary epithelialization (secondary intention). By 1872, despite their description as epidermic grafts, Reverdin and Pollock had both admitted that their grafts contained a portion of dermis.

Around the same time (1870), the British surgeon George Lawson (1831–1903) presented to the Clinical Society of London his experience using a full-thickness skin graft from the upper arm to repair a complete ectropion of the upper eyelid. Lawson’s method described meticulous dissection of all subcutaneous tissues from the dermis. Due to its size of only 32 mm, his grafts were affectionately referred to as “Fourpenny Grafts” [33]. Despite his accomplishment, however, Lawson is largely overlooked by historians. It would not be until 1875 that Polish-born John Reissberg Wolfe (1824–1904), an ophthalmologist from Glasgow (United Kingdom), would describe his experiences in the *British Medical Journal* that the medical community would take notice. Wolfe used even larger grafts (2.5 cm × 5 cm) from the forearm to reconstruct lower eyelid ectropions in a manner much like Lawson’s. Then, 20 years later (1896), Fedor Krause (1857–1937), a neurosurgeon in Hamburg (Germany), emulated the same practice as Wolfe in a larger series. In 1893, he recommended the use of Wolfe’s techniques to the 23rd *Kongress der Deutschen Gesellschaft für Chirurgie* (Congress of the German Surgical Association) and highlighted the specific instances where predecessors had failed. He also pointed out the benefits of full-thickness skin grafts over thinner grafts: resistance to scar contracture, improved joint range of motion, and more favorable aesthetic results. For his efforts Krause is credited with popularizing full-thickness skin grafting [12]. Although neither Wolfe nor Krause can reasonably be credited for discovering full-thickness skin grafts, their contributions to popularizing the technique, which is ubiquitously used today, is why the dermo-epidermal graft is still remembered under the eponym “Wolfe–Krause graft” [34].

The evolution of the split-thickness skin grafting technique follows a parallel timeline. Certainly, Reverdin’s pinch grafts were historic in the origin of skin graft of all degrees of thickness. In his initial description of the grafts, Reverdin suggested that they were composed almost entirely of epidermis. It was not until later that he would admit to some dermal inclusion. In 1872, the French surgeon Leopold Louis Xavier Édouard Léopold Ollier (1830–1900) of Lyon (France) published his experiences with thin skin grafts ranging from 4 to 8 cm^2^ in size, substantially larger than previously performed by Reverdin or Pollock [5]. Ollier’s grafts, though described in the same nomenclature as full-thickness skin grafts at the time, were considered intermediate thickness as they included the entire epidermis and only a thin portion of dermis. Today, we would consider Ollier’s grafts to be split-thick skin grafts (STSG). Due to the technologic limitations at the time, the collection of such grafts required remarkable technical dexterity [5,12]. Ollier’s grafts were, however, more advantageous over full thickness grafts in two remarkable ways. The first is that the grafts experienced less shrinkage and curling at the time of collection due to the decreased amount of dermal elements. The second is that the donor site required far less time to heal [14]. 

Carl Thiersch (1822–1895), a German surgeon who trained under Dieffenbach, believed himself to have perfected skin grafting with his technique, which involved minimal dermal inclusion in the graft. He demonstrated his method to the 15th Kongress der Deutschen Gesellschaft für Chirurgie in 1886, and for a time his grafts were referred to as “razorgrafts” [22]. Critics attest that Thiersch’s conclusion that the “ideal skin graft” should be excised as superficially as possible reinforced the incorrect notion that there could only be one optimal graft thickness, and that such idealism delayed the development of modern full-thickness skin grafting techniques. Thiersch emphasized his opinion in a number of his own publications and was broadly influential within the medical community due to his reputation as a brilliant surgeon [35]. Today, we recognize that there is no ideal thickness for skin grafts. Rather, grafts of different thicknesses can be used in a diverse variety of applications. Furthermore, despite not ascribing any credit in his writings to Ollier, who published similar findings 14 years earlier, both are credited by historians for their advancement of surgical knowledge and technique. The Ollier–Thiersch graft is now synonymous with STSG. Despite Thiersch’s emphasis on meticulous technique, grafts performed by his contemporaries would, progressively over time, get thicker. Eventually, inclusion of nearly half the dermis was still considered an Ollier–Thiersch graft. This distinction would be obviated with advances in surgical technology in the twentieth century.

Thiersch was in many ways was obsessed with the microscope, taught lectures on microscopic anatomy and was even described as a “dexterous microscopist” in his obituary, for his integration of his passions for cellular physiology and surgery. Thiersch is reported to have rigorously studied the microvascular anastomosis between skin grafts and wound beds. He described layers in the granulation tissue that formed, distinguishable by the orientation of the vasculature. Through his study, he discovered that superficial granulation tissue actually hinders skin graft implantation. 

“The only alternative, therefore, is elimination of the superficial portion of the granulation (layer) and implantation of the skin (graft) directly upon the tense underlying tissue. This substratum is exposed by sharp, horizontal incision, hemorrhage is permitted to run its course completely, and then skin graft is placed upon this greyish wound surface, whose vessels and tissues are in excellent condition for immediate inflammatory adhesion.”—Carl Thiersch [22].

Thus, through the efforts of countless surgeons, the basic tenants of skin grafting from epidermal to full-thickness grafts, as well as the optimal preparation of the wound bed were established by the end of the nineteenth century.

## 6. Skin Grafts in the Contemporary Era—World War II and After: 1939–2020 AD

### 6.1. Burn Wound Management

The use of skin grafting in acute burn wound treatment did not occur until the utility of early wound excision in full thickness burn wounds was understood. Although the importance of early burn wound excision was first described by Paré in the sixteenth century. Wilhelm Fabricius Hildanus (1560–1634) is considered the “Father of German Surgery” and the first to write a book dedicated to the management of burn wounds, De Combustionibus (On Burns) in 1607. In his book, Hildanus advocated for the surgical removal of burn eschar to facilitate improved medication penetration. Although certainly ahead of his time in a surgical regard, Hildanus was ignorant of the teaching of Paré nearly 70 years prior, as many of the medications he utilized were of medieval origin [32]. In the eighteenth century, Dieffenbach would describe the use of skin grafts to reconstruct wounds caused by burn injuries and in the late nineteenth century both Reverdin and Pollock would be credited with the successful use of skin grafts in chronic burn wounds. However, it would not be until the 1940 that skin grafting following tangential excision of acute burn wounds wound be connected in a therapeutic sequence. 

The 1940s was a period of significant advancement in the understanding of burn shock management, not in any small part due to the immense loss of life precipitated by multiple man-made disasters. In 1921, Frank Pell Underhill (1877–1932) a surgeon at Yale University, treated more than 20 victims of the infamous Rialto Theater fire in New Haven, Connecticut. Underhill noted similarities in the serous fluid within skin blisters and plasma, and went on to suggested that acute shock in burn victims was primarily a hypovolemic process due to fluid losses from injured skin [36]. Then, 20 years later, Oliver Cope (1902–1994) and Francis D. Moore (1913–2001) from Massachusetts General Hospital would treat nearly 40 victims of the Coconut Grove Nightclub in Boston, Massachusetts. Their work would connect the amount of body surface injured to the volume of resuscitation fluid needed to stave off the precipitating shock [37]. In 1942, Forrest Young of the University of Rochester correlated that victims of burn injuries suffered from sepsis and shock as a result of fluid losses and bacterial colonization of their wounds. He also surmised that full-thickness burn injuries would only heal by secondary intention. Therefore, he advocated for early excision and skin grafting to improve mortality by removing the source of sepsis [38]. During this era, however, many believed it inopportune to operate during the acute period of shock, so the term “early surgical intervention” referred to a period of 10–21 days after the injury [39,40]. 

In 1960, Douglas MacGilchrist Jackson (1916–2002) and colleagues from Birmingham, England described in detail a series of cases where early excision down to fascia was performed on the day of injury for full thickness burn wounds and hemodynamic management with aggressive hemoglobin monitoring and transfusions as needed. They determined that 20–30% total body surface area (TBSA) burns could be operated on the day of injury without any increased risk of death, while achieving much earlier rates of graft take and wound closure [41]. Other surgeons would advocate for similar time frames for early excision of burn wounds, but there was little momentum in the surgical community because like Jackson, they could not demonstrate a mortality improvement over the method that delayed excision by 2–3 weeks. This would change in the early 1970s when an unknown Eastern European surgeon, Zora Janžekovič (1918–2015) from the Slovenian city of Maribor, published her findings after performing tangential excisions on deep-second and third-degree burn wounds on 2615 patients [42,43]. Janžekovič described the austere post-war conditions in which she found herself practicing in isolation. Later in her life, Janžekovič wrote:

“The daily changing of dressings of the burn patient, piles of dressings full of pus, the terrible stench, but above all the horrible suffering of the patients—mostly children who were scared to death and emaciated, was a cry for help and a challenge for our personal engagement. Their suffering became our suffering. The feeling of our own helplessness and the incompetence of the then medical science were destroying us… Confronted with this terrible situation, I was forced to search for any kind of solution.”—Zora Janžekovič [44].

She went on to abandon delaying wound excision until the wound had fully demarcated through the sequela of infection out of necessity. She believed that she could circumvent pathologic process by shaving the wound down to healthy tissue before infection had set in. Janžekovič found that tangential excision needed to be performed down to bleeding tissue, otherwise any applied graft would desiccate and fail. Beginning with small wounds and gaining confidence with larger injuries, Janžekovič is credited with formalizing the technique of tangential excision with immediate skin graft placement within 5 days of injury. Her results showed that patients with up to a 20% TBSA could be healed within 10 days, which was unheard-of at other more prestigious burn centers [44]. Janžekovič would go on to be the first woman to receive the Evens Medal from the American Burn Association and the Zora Janžekovič (Golden Razor) Award from the European Club for Paediatric Burns. It should be noted that Janžekovič technique was primarily for deep second-degree burns that were small enough to be addressed with a single operative intervention. In the decades that followed her publication, more than 200 surgeons from around the world would come to learn her technique first-hand and go on to further build upon her accomplishments. John Francis Burke (1922–2011) and his colleagues from Harvard University showed that the combination of tangential excision for smaller burns and full-fascial excision for larger burns followed by immediate autograft placement markedly reduced mortality and allowed the successful treatment of children up to 80% TBSA [45]. In the decades since, numerous surgeons continue to validate and build upon Janžekovič’s technique. Today tangential excision of all non-viable tissues within 72 h of injury followed by immediate skin graft placement for full-thickness wounds is considered the standard of care.

### 6.2. Operative Equipment

Prior to the nineteenth century, skin grafts were collected using scalpels and knives adapted for surgical procedures, such as the Catlin knife (also called the Catling knife, Amputation knife, and Interosseous knife), which was a double-bladed instrument that was typically 17 cm long and 1.5 cm wide with a simple handle at one end, as seen in Figure 1a [12]. As a basic single-piece instrument, it lacked any mechanism by which to control depth of excision. It was also used surgically for a number of different tasks [12]. Although commonly used as far back as the seventeenth century, the Catling knife is still used by many surgeons today when performing extremity amputations. As one might imagine, the fine task of harvesting skin grafts with such a blunt instrument was challenging even in the most experienced hands and generally resulted in inconsistent graft thickness.

In 1920, the Thiersch’s skin grafting knife was introduced and was a rectangular stainless steel single-bladed instruments weighted toward the handle. This rectangular design, as seen in Figure 1b, has persisted with nearly all subsequent hand-held skin grafting instruments. Vilary Papin Blair (1871–1975) an American surgeon at Washington University in St. Louis, who is best remembered as a pioneer for helping to distinguish plastic surgery from general surgery. Much of his reconstructive experience came from his time as a surgeon in the United States Army during World War I. He introduced the Blair knife in 1930, which was used in conjunction with a suction apparatus that put the skin under tension during harvest. This markedly improved the consistence of the grafts and allowed surgeons to harvest skin free-hand [12]. The Hofmann and Finochietto knives were introduced shortly thereafter, both of which were equipped with rudimentary guards that could be adjusted with lateral screws to allow the surgeon to calibrate the depth of dissection.

Improving upon this design, Thomas Graham Humby (1909–1970) a British plastic surgeon training at the Great Ormonde Street Hospital for Sick Children (London, UK) under the famous Sir Heneage Ogilvie introduced the Humby knife in 1934, as seen in Figure 2a [46]. Humby added a guard with a roller mechanism to his knife that allowed detailed calibration of the depth of tissue excised. In addition, the Humby knife had a rectangular metal frame equipped with 1/8th inch hooks at either end and a ratchet mechanism that enabled the surgeon to keep the donor tissue under tension while sliding the knife within the construct of the frame. Its use revolutionized skin grafting, enabling surgeons to single handedly and consistently excise rectangular strips of skin of consistent depth. The framework of the Humby knife was later abandoned as it could not reasonably be used for a number of potential donor sites. Subsequent variations of the Humby knife design include the Modified Humby (1936, fixed blade for rigidity), Bodenham (1949, partially supported replaceable blade that must be dismantled to change blades), Braithwaite (1955, leaf-type fully supportive replaceable blade that can be changed without disassembly) [47,48,49]. All of these knives shared two design flaws. First, the mechanism for depth adjustment is dependent on two separate knurled collars mounted on either end of the back of the handle allowing room for asymmetry and user error. Second, the roller guard and the handle are fixed while the blade must have some slack in the end-bearings in order for it to slide freely from side-to-side during use. As a result, grafts would often curl around the guard, become irregular and at times became gradually thicker across a harvest.

John Watson (1914–2009), who served as a pilot in the Royal Air Force during World War II, went on to become a plastic surgeon at the Queen Victoria Hospital in East Grinstead, United Kingdom, after the war. There, he would operate on many former aircrew who had sustained disfiguring burn injuries during the war. Inspired by a potato peeler and wanting to create a user-friendly instrument that emulated the renowned dexterity of his mentor Sir Archibald McIndoe (1900–1960). So inspiring was McIndoe’s work that he was knighted for his contributions in reconstructing injured veterans of World War II. Watson introduced his knife in the year of McIndoe’s passing, 1960. The Watson knife, as seen in Figure 2b, is unique in its simple design with a single more ridged knurled control knob for depth adjustment and precision-fit end-bearings that do not slide [50,51]. The result was a knife that is easier to operate and maintain, and for this reason the Watson knife is still used at many burn centers today.

In 1937, a remarkable advancement in surgical technology occurred with the introduction of the Padgett–Hood dermatome. Earl Calvin Padgett (1893–1946), an American physician from Kansas City who trained under Blair during residency. Padgett in collaboration with his colleague George J. Hood from the Department of Engineering designed a simple to use dermatome that allowed calibrated collection of skin grafts and presented their invention at the Western Surgical Association in 1938. Unlike many of its predecessors, the Padgett–Hood dermatome earned instant recognition. One of the founding members of the American Board of Plastic Surgery, George Warren Pierce, called it “the greatest contribution in many decades to the technique of skin grafting” [52].

The Padgett–Hood dermatome consisted of an aluminum drum that utilizes a traction-adhesion principle to feed donor tissue into a rotating blade, as seen in Figure 3a. The distance between the blade and the rotary drum can be calibrated down to a thousandth of an inch. The Padgett–Hood dermatome had several advantages compared to previous free hand tools. It vastly improved the quality and consistency of harvested graft, even when utilized on uneven donor surfaces. This not only increased the body surface that could be considered for donor harvest, but also increased the pool of surgeons that could perform skin grafting. No longer were skin grafts an art restricted to the most dexterous and experienced plastic surgeons, rather, now they could be performed by nearly any surgeon in training. Padgett’s fortunate timing just prior to the onset of World War II cannot be understated [29]. Blair Rogers claims the Padgett–Hood dermatome “probably did more than any other single achievement in our specialty to bring the advantages of rapid, free skin transplantation to the innumerable casualties of that conflict” [31].

Following the war, John Davies Reese (1893–1958) would improve upon the Padgett–Hood dermatome in 1946. In contrast to the cast-aluminum of its predecessor, the Reese dermatome was a heavily machined, more precise and more reliable instrument. Notable disadvantages of the Reese dermatome were its considerable weight and inability for depth to be adjusted during a graft harvest [53]. The Reese dermatome would be overshadowed by the introduction of the first electric dermatome by the American surgeon Harry M. Brown (1914–1948) in 1948, as seen in Figure 3b [12]. Brown’s dermatome was hand-held and relatively easy to operate, which allowed the harvest of large amounts of skin graft rather quickly with minimal effort. Brown actually thought of this new instrument while being held prisoner by the Japanese during World War II, but unfortunately was killed in a tragic accident shortly after his invention was introduced. A number of contemporary electric dermatomes including the Stryker, Padgett, and the Zimmer are based directly on the design of the Brown electric dermatome [12].

### 6.3. Skin Graft Expansion

Split thickness skin grafts first became popular in the 1930s after Blair and James Barrett Brown (1899–1971) first articulated the differences between full-thickness, intermediate-thickness and epidermal skin grafts. Their work showed that donor sites for STSG healed through epithelialization from local hair follicles and the underlying basal layer. The preservation of donor tissue so that it could potentially be re-harvested made STSG an attractive option over full thickness grafts, particularly in the treatment of large surface area wounds. The invention of the dermatome made the collection of STSG common practice to any interested surgeon. Building upon these advancements, Cicero Parker Meek (1914–1979) from South Carolina introduced a novel micrografting technique in 1958 that enabled a graft to be utilized over a wound ranging from six to nine-fold greater in size [54]. Meek built upon the early wisdom of Reverdin’s pinch grafts and Lawson’s Fourpenny grafts by recognizing that epithelialization occurred from the graft edge. He hypothesized that by maximizing the epithelialization boarder he could expedite wound healing. Therefore, Meek would cut each square inch of harvested graft in a 16 by 16 grid pattern (generating 256 micrografts each 1/16^th^ of a square inch) using the Meek–Wall microdermatome. The result increased the epithelialization boarder 16-fold from 4 inches to 64 inches. The micrografts would then be placed onto bandages and applied to the wound [55]. In addition to the expansion of applicable surface area for harvested donor tissue, the Meek’s technique allowed serosanguineous fluids to drain freely around the micrografts. Although the Meek–Wall microdermatome is still utilized at many specialized burn centers today, its routine use did not gain momentum as it was expensive to acquire and cumbersome to operate.

Although the concept of micrografting would not be forgotten, at this time in history the Meek’s technique served as the bridge to skin meshing, which also allowed for wound fluid drainage and expansion of donor tissue over a larger wound surface area. Credit for the first prototype skin mesher belongs to the Swiss surgeon Otto Lanz (1865–1935) in Amsterdam (Netherlands), who trained under the Emil Theodor Kocher (1841–1917) the first surgeon to be awarded the Nobel Prize. Lanz invented a tool in 1907 he called the Hautschlitzapparat, which made 19 parallel cuts into a harvested skin graft, allowing it to expand in a manner that he described as a skin-net [56]. The Hautschlitzapparat and the accompanying tissue pattern are depicted in Figure 4a. Lanz described that the meshed skin graft would allow it to cover twice the surface area as the original donor tissue, a principle called the concertina effect. Lanz would apply one-half of the graft over the wound of interest and the other half would be used to re-cover the donor wound. Lanz’s idea was based on the children’s activity where a strip of paper is cut in alternating intervals to make an accordion-toy. He did this because it bothered him that when using Thiersch grafts often the wound would heal prior to the donor site. In 1930 Beverly Douglas (1891–1975) described the “sieve graft”, which was a skin graft where he punched out holes. The removed “holes” were then re-applied on the donor site, while the graft was applied to the wound. The intention of the sieve graft was to facilitate drainage of wound fluids, which if undrained can dissociate the graft from the underlying wound bed and compromise viability. In 1937, multiple surgeons like Lester Reynolds Dragstedt (1893–1975) and František (Francis) Burian (1881–1965) continued to adapt and modify the sieve graft by manually making staggered incisions in the graft instead of hole-punches. As a result, fluid drainage was still achieved while the graft could now expand to cover larger surface areas. Although at that time, this technique was used primarily on full-thickness skin grafts, today it has been adapted for STSG and is commonly referred to as “pie-crusting”.

In 1964 James Carlton Tanner Jr. (1921–1996) and his chief resident Jacques J. Vandeput from Grady Memorial Hospital in Atlanta, Georgia created a simple device called the Tanner–Vandeput mesh-dermatome that could expand STSG by a ratio of 1:3. The Tanner–Vandeput mesh-dermatome and its accompanying tissue pattern are depicted in Figure 4b. In their landmark paper titled “The Mesh Skin Graft” they introduced the term “meshed graft” [57]. The Tanner–Vandeput mesher consisted of two four-inch rollers, one knurled to grip the skin and the other with multiple parallel staggered cutting blades to cause the meshing pattern. Harvested skin grafts were placed between the two rollers and then rotated to incise a meshed pattern into the graft [58]. As mentioned, however, micrografting would not be forgotten. In 1966, Vandeput combined the concepts of the Meek–Wall dermatome and the Tanner–Vandeput mesher to create “ultra-postage skin grafts” that were 1/20th of an inch squared [59].

The Tanner–Vandeput mesher was originally sold under the commercial name Mesh-Dermatome I by the Zimmer Company (Dover, OH, USA) in 1964 and utilized a plastic feeding tray called a dermacarrier. A number of iterations have since been made over the years to improve upon the design. The Mesh-Dermatome II (Zimmer Company) introduced in 1970 changed the blade angles relative to the dermacarrier and the meshing pattern (hexagon) to allow variability in the meshing ratio from 1:1 up to 9:1. In 1991, the Zimmer Skingraft Mesher (Zimmer Company) utilized a ratchet and cog-wheel mechanism to pull the dermacarrier through the cutting mechanism. This model also allowed interchangeable bladed rollers to allow for rapid variation in ratio (ranging from 1:1 to 4:1). The two companies did away with the dermacarrier and introduced a double-cutting-roller design—Collin (Arcueil, France) in 1986 and Brennen Med (St. Paul, MN, USA) in 1988. The primary benefit of the double-roller model is that it did not require sharpening as it did not relay on blades piercing the skin graft in order to form interstices. Rather, the two rollers, one of which is notched, performed a scissor-like pinching action. The Brennen Skingraft Mesher (Brennen Med) which was modified in 1993 from the original design to be more user-friendly offers a number of meshing patterns, but, unlike its Zimmer counterpart, a different instrument is needed for each meshing ratio (ranging from 1:1 to 8:1) [60]. Meshed grafts are the mainstay of modern burn wound care and use nearly universally by all burn centers. They have several key advantages over unmeshed (sheet) grafts beyond surface area expansion, fluid drainage, and expedited wound closure. Meshed grafts are more versatile as the interstices allow the shape of the donor tissue to be adapted to asymmetric wounds and across irregular body contours [58]. Unmeshed skin grafts in contrast develop more robust vascularization, reduced scar contracture and are more aesthetically pleasing due to the lack of interstices.

Severely injured burn victims and limited donor site availability have continued to challenge burn surgeons to push the limits of skin graft expansion. In 2012, Florian Hackl and colleagues from Brigham and Women’s Hospital in Boston, Massachusetts described the Xpansion^TM^ technique, which afforded an expansion ratio of 100:1 by mincing skin grafts into 0.8 mm × 0.8 mm micrografts and then spreading them onto the wound bed [61]. Studies have found that mincing grafts leads to over expression of growth factors like tumor necrosis factor alpha, platelet-derived growth factor, and basic fibroblast growth factor, which are thought to promote wound healing [62]. Further, developing Hackl’s technique, Denesh Kadam an Indian surgeon from Karnataka, India developed a technique called “Pixel Grafting”. Similar to the Xpansion^TM^ technique, skin grafts are minced into digital pixel sized grafts (0.3 mm × 0.3 mm) and are able to achieve expiation ratio of up to 700:1 [63].

### 6.4. Homografts and Immunologic Discoveries in Skin Grafting

When Reverdin first introduced the concept of skin grafting it gained popularity rapidly due to the remarkable nature of the short-term results. By the 1870s, however, Reverdin’s techniques were losing popularity because of their less impressive long-term results. From our modern perspective it is not intuitive why this might be. At the time these shortcomings were attributed to impracticality of the concept of skin grafting itself, but in fact the reason had to do with the unintentional introduction of homografting—the use of skin grafts across members of the same species. Reverdin, along with numerous other surgeons of his age, held the conviction that homografts were for all intents and purposes interchangeable with autografts. In fact, in many publications from that period the authors failed to bother specifying what type of graft was even used. Reverdin’s reputation in burn surgery hinges on reports that he used his own skin to treat injured patients. In 1872, he wrote about the care of a burn victim:

“In my first grafts I used skin from the patient himself, but I soon became convinced that the results was the same when I used skin from another individual. This has been demonstrated with certainty.”—Jacques-Louis Reverdin [64].

Although the idea of a surgeon using their own skin seems remarkably philanthropic by our modern cultural norms, during Reverdin’s time many surgeons would excise fragments of their own skin to demonstrate to patients who were particularly scared that the procedure was in fact not as painful as the patient might imagine. The modern surgeon might even equate it to a form of informed consent. Surgeons also reported that many patient’s relatives were more than willing to donate fragments of their own skin to aid in the healing of a loved one [65]. Going one step further, in the Berlin Military Hospital a skin graft donor could be found for as little as the price of a beer [66]. Donor tissue were also collected in more creative ways—for example from amputated extremities and circumcised foreskin [67,68]. The timing between graft harvest and placement, as well as the temperature that the collected tissue was stored at, was a matter of great debate in those days. Some surgeons arguing that the procedure needed to occur as soon as possible and that the graft must be kept warm, while others reported success with grafts that had been collected four days prior and were stored at 10 °C [69,70]. This dichotomy resulted in a number or remarkable scenarios. One surgeon described placing the soon-to-be amputees in the same operative theater adjacent to the anticipated skin graft recipient. Another surgeon described his method of keeping grafts warm by placing them in his armpit while transporting them from donor to recipient [70,71]. In addition, despite amputations being routinely performed in those days, disease like smallpox, syphilis and tuberculosis were also equally commonplace. Therefore, the sudden appearance of numerous cases of infectious diseases transmitted through homografts in the literature should come as no surprise to the modern reader [64,72,73]. Surgeons also described progressive graft degradation and failure in a time when the mechanism of graft rejection was not understood. Therefore, for a time homografting was abandoned by most practitioners, and considered to be an inherently unsuccessful enterprise.

In the 20th century, however, with advances in critical care and medical technology, there was an unexpected revival of cadaveric homografting with the introduction of the skin bank and cryopreservation. In 1903, the German physician Johann Wentscher (1852–1913), a contemporary of Thiersch, was the first to document the viability of refrigerated skin grafts at 0 °C for 14 days [74]. Long-term cryopreservation was not possible until the 1930s when effective, reproducible, standardized storage methods were established. It was during World War II that serious investment went into skin banking due to the large number or wounded soldiers that were returning from the war. The sheer burden of injury produced by World War II spurred the development of skin banks and trauma-specific research centers, such as the United States Institute of Surgical Research [75]. Researchers found that harvested skin could be preserved in a glycerol-based cryopreservative for up to up to four months at −79 °C [76]. Such skin banks allowed wartime hospitals to have a read supply of homograft skin for severe burn victim with insufficient donor site.

Around the same time, across the Atlantic, the Battle of Britain (1940) raged and a plane crashed in Oxford near the home of British immunologist and Zoologist, Sir Peter Medawar (1915–1987). The physician caring for the horribly burned pilot consulted Medawar for advice. Although Medawar had no experience caring for burn victims, he believed that skin grafting would afford the airman the best chance of survival. Unfortunately, despite Medawar’s efforts, the pilot would not survive. The experience, however, would instill in Medawar a life-long curiosity about the immunological intricacies of skin grafting. During the remainder of the war Medawar collaborated with the Scottish Plastic surgeon Thomas Gibson (1915–1993) to perform homografts and autografts on other soldiers [77]. The two observed that homografts, though initially appearing to incorporate, would go on to be rejected within two weeks’ time. In contrast, autografts were often successfully engrafted during the same time frame. Medawar also noticed that if a second homograft from the same donor was re-attempted, graft rejection occurred more quickly. This affirmed Medawar’s suspicions that the etiology of graft failure was immune-mediated. He reported his findings to the War Wounds Committee of the Medical Research Council in 1944, The Behaviour and Fate of Skin Autografts and Skin Homografts in Rabbits [78].

In 1945, Ray David Owens (1915–2014) introduced the concept of chimerism while studying dizygotic cattle twins. Owens observed the presence of “mixed blood types” that were the result of in utero genetic exposure [79]. Frank Macfarlane Burnet (1899–1985) an Australian Virologist built upon these findings and proposed the theory of immune tolerance, suggesting that immunologic self-awareness could be influenced, particularly during embryogenesis. Medawar tested this theory by crossing allografts between dizygotic cattle twins. He observed that grafts remained intact for several weeks longer than would typically be expected. Medawar took the experiment one step further using a mouse model. He inoculated fetal mice with splenic cells from a donor (second) mouse strain. After eight weeks, he performed allografts using the donor strain of mice and observed that the transplanted skin was tolerated. As a control, skin from a previously unexposed (third) strain was also grafted and was expectantly rejected [80]. This experiment is credited as the foundation for modern transplant immunology and both Medawar and Burnet were awarded the Nobel Prize in Medicine for their discovery of immune tolerance.

Prior to World War II, Colonel James Barrett Brown (1899–1971) had postulated that homograft rejection was due to the genetic disparity between donor and host. In 1937, he performed the first successful “homograft” in which both the donor and recipient of a skin graft were identical twins. During the war as the Chief of plastic surgery at Valley Forge General Hospital (Phoenixville, Pennsylvania), Brown took Joseph E. Murray (1919–2012), then a surgical intern and First Lieutenant, under his wing. This act of charity spared Murray overseas deployment and afforded him experience caring for wartime victims of burn-related trauma. Similar to Medawar, Murray’s first-hand exposure to burn victims would inspire an academic career in tissue transplantation. Murray would lay the foundation for our modern understanding of skin’s enhanced antigenicity [81,82]. In the decades that followed World War II, a significant amount of research on the immune response to skin grafts was performed. In the 1950s, major histocompatibility complexes (MHC) were discovered. However, Rupert Everett Billingham (1921–2002) demonstrated that both MHC-matched and mismatched donor homografts resulted graft failure [83]. The 1960s saw a wave of animal experiments with immunosuppressants like phenothiazine derivatives, methylhydrazine derivatives, anti-lymphocyte biologics, steroids, anti-metabolites, and even x-ray irradiation—none of which were effective in reducing skin graft rejection rates or were practical for use in human use [84]. In the 1970s, transplanted skin was found to generate a more robust immune response than solid organs due to its higher antigenicity. Although the exact mechanisms for cell-mediate (T-cell) and innate (Natural Killer cell) mediated acute rejection was not better understood till more recent decades, surgeons have come to understand that homograft do not replace the need for autograft, but serves as temporary bridge to autograft application. Of note, Murray replicated the work of him predecessor, Brown, performing the first living donor kidney transplant between identical twins, and was awarded the Nobel Prize in Medicine for his achievement in 1990 [81].

In 1954, Douglas MacGilchrist Jackson (1916–2002), a contemporary of William Heneage Ogilvie, was a British surgeon from the Birmingham Accident Hospital who is remembered for his “alternate strip method,” which involved placing half-inch strips of autograft and homograft in an alternating sequence to cover large posterior thoracic burn wounds. Jackson credited the idea to Rainsford Mowlem (1902–1986), a New Zealander who was appointed to presidency of the British Association of Plastic Surgeons. Jackson performed this method on 16 patients and reported successful results with many patients being able to return to their lives without significant functional disability within a year. Jackson also coined the term “creeping substitution”, an observation he noted when epithelialization from the autograft strips boarders would grow underneath the homograft strips and eventually connect adjacent autograft strips. Subsequently, the strips of homograft would separate and reveal an epithelialized wound bed underneath [85].

In 1986, Ming-liang Zhang, a Chinese surgeon from Beijing Jishuitan Hospital, and colleagues further developed this technique by slicing autografts into one millimeter micrografts, which they would place onto a larger sheet of homograft. They called this autograft–homograft composite “Microskin”. Sheets of Microskin were then applied to the wound bed and much like in the alternate strip method, autograft epithelial cells proliferated via creeping substitution and eventually separated from the homograft. Over multiple publications, Zhang described the successful use of Microskin grafting in 32 burn patients (ranging from 2.5 to 45% TBSA) and reported expansion ratios as high as 15:1 to 18:1 [86]. Despite its advantages, however, Microskin remains technically challenging and requires specialized equipment to apply routinely. J. Wesley Alexander (1934–2018) and colleagues from the Shriners Burns Institute in Cincinnati, Ohio (now called Shriners Children’s Ohio) utilized a method in which 6:1 meshed autograft was reinforced with 3:1 meshed homograft. Alexander noted that by layering homograft on top of the more friable autograft the latter would be protected during the critical period of incorporation. In the 14 patients this method was initially described in, there was a 99% graft take of the underlying autograft with no reported loss at follow-up. The overlying homograft, in contrast, had a 95% initial incorporation within the first three days and a subsequent near-total rejection over the subsequent 30-day period [87]. The Alexander method or, as it is more commonly called, the “Sandwich technique” has been modified since being introduced, but is still widely utilized today at almost every large burn center today for its simplicity and efficiency.

### 6.5. Skin Substitutes

Bioengineered skin substitutes have also emerged as an area of interest. Research into skin substitutes dates back to 1975 when Ioannis V. Yannas of the Massachusetts Institute of Technology (MIT) and John Francis Burke (1922–2011), the Chief of Staff at Shriners Burns Institute in Boston (now called Shriners Hospital for Children©—Boston) and the Massachusetts General Hospital Burn Service, collaborated to develop the first bio-synthetic skin substitute called Integra^®^ (Integra LifeSciences Corp., Plainsboro, NJ, USA). In recognition for their achievement, Yannas and Burke were inducted into the National Inventors Hall of Fame in 2015. Integra^®^ consists of a layer of cross-linked bovin collagen and shark chondroitin (glycosaminoglycan) with a silicon top-layer. The collagen-chondroitin matrix facilitates the recapitulation of a reticular dermis, while the silicone acts as a temporary protective pseudo-epidermis. After the excision of a full-thickness burn, Integra^®^ allows the wound bed to regenerate a layer equivalent to the dermis. After approximately three weeks, when the dermis has regenerated, the silicon layer is physically removed and replaced with a standard autograft [88]. Products like Integra^®^ are acellular skin substitutes. In contrast, cellular skin substitutes like Transcyte^®^ (Advanced Tissue Sciences, La Jolla, CA, USA) consist of a synthetic scaffolding seeded with living human cells (fibroblasts).

Cell-based therapies aim to replace lost tissue with cultured skin cells, which was not considered feasible until 1975, when Howard Green (1925–2015) and his graduate student James G. Rheinwald at MIT successfully cultured human keratinocytes [82]. Green and Rheinwald actually made their discovery by accident while they were trying to replicate a teratoma, an altogether different tumor [89]. Their discovery called Cultured Epithelial Autografts (CEA) involves harvesting stems cells from the patient’s skin and growing a culture of these cells into an autograft sheet, which can then be applied to burn wounds and was particularly useful when donor sites are limited. In 1983, the mettle of CEA was put to the test when a five-year-old, Jamie Selby, and his seven-year-old brother, Glen Selby, suffered 97% TBSA third-degree burns after playing with flammables in an abandoned building. Both children from Casper, Wyoming were airlifted to Shriners Hospitals for Children©—Boston where Green and his colleagues performed over 350 grafts grown from small patches of donor tissue. Having only produced CEA on a small scale prior to this occasion, Green restructured his laboratory at Harvard University in order to produce CEA around the clock for the two boys. He later went on to found the company BioSurface Technology Inc. (Cambridge, MA, USA). The survival of the two boys, gained CEA national recognition from the medical community. In 2010, Rajiv Sood and his colleagues from Indiana University in Indianapolis, Indiana described their experience with the use of CEA in 88 victims of major burns, ranging from 28 to 98% TBSA over a period of 18-years. They found an overall graft success of 72.7%, and a survival rate of 91%. Sood wrote that “[such results] gives much optimism for continuing to use CEA in critically burned patients” [90]. Despite its significant benefits, the primary barrier to its routine use in the critically ill burn patient is the two to three weeks of incubation-time needed to produce it.

In 1989, Steven T. Boyce from the Shriners Burns Institute in Cincinnati, Ohio and colleagues from the University of California San Diego Medical Center introduced an alternative product which they expected to replace CEA called Cultured Skin Substitutes (CSS). Consisting of a collagen-glycosaminoglycan sheet inoculated with human dermal fibroblasts on one side and epidermal keratinocytes on the other, CSS could potential to preserve donor sites as it did not rely on autologous cell harvesting [91]. Boyce prodigiously published the benefits of CSS, showing in vitro that it could be modified in various ways to emulate the anatomic features of autografts—“lipid supplements” could enhance the epidermal barrier, pigmentation could be optimized to the patient with the addition of melanocytes, the addition of growth factors could enhance wound healing and the CSS itself could be genetically enhances to expedite wound closure [92,93,94,95]. For a time, CSS was used at a number of burn centers, especially within the Shriners system, however, it did not gain sufficient traction from the Food and Drug Administration to be adapted for widespread clinical use.

Around the same time, another emerging technology was gaining attention within the tissue translocation community—cellular suspension. Rupert E. Billingham (1921–2002) and Joyce Reynolds of the University College (London, United Kingdom) first introduced cellular suspension in 1952 using an enzymatic preparation with Trypsin [96]. Although the suspended cells were viable, this early technique was not successful because the cells failed to adhere to the wound bed. In 1988, János Hunyadi and colleagues revisited the technique by trypsinizing keratinocytes and suspending them in a fibrin glue solution, a process which they named Keratinocytes in Fibrin Glue Suspension (KFGS). Hunyadi was able to use KFGS technology to successfully treat venous stasis ulcers, but did not apply it to the treatment of burn wounds [97]. In 1994, Hans-Wilhelm Kaiser and colleagues from the University of Bonn Medical School and Burn Centre in Colonge-Merheim, Germany published the use of fibrin glue suspension techniques with cells derived from CEA in the treatment of burn wounds [98]. Several subsequent studies compared and contrasted aerosolized skin cells with and without fibrin glue, and, in 2003, Lachlan J. Currie and colleagues concluded that there was no observable of histologic difference in outcomes [99].

The merger of autologous epithelial cell culturing and cellular suspension occurred in the early 1990s by the British-born Australian plastic surgeon, Fiona Melanie Wood. As the Burn Director of the Royal Perth Hospital, Wood had experienced first-hand the morbid consequence of awaiting CEA production in a critically injured burn patient. After several deaths, which Wood believed could have been salvaged if the production time of CEA could be decreased, she began experimenting with the production process of cultured epithelial cells. In her early work, Wood discovered that CEA could be applied after culture growth of as little as 10 days. Not only would the sub-confluent cultured graft adhere to the wound, but Wood observed that the wound healed quicker. The mechanism for this observation was unknown, but Wood suspected that it was a result of cellular signaling pathways within the wound bed that promote proliferation or that sub-confluent cells inherently had improved proliferative potential having spent less time in culture medium [100]. Edwin A. Deitch and colleagues also found that burn wounds closed after 21 days, independent of the method, incurred a 70% risk of hypertrophic scar formation, while wounds closed within ten days only afforded a 4% risk [101]. Wood recruited Marie L. Stoner, a cell biologist, to help her reduce cultured graft application time table and the two worked tirelessly—successfully minimizing it to 5 days. In their method, a postage-stamp size piece of uninjured donor epidermis is harvested. Keratinocytes are scrapped off and cultured in a concentrated single-cell suspension for 5 days and collected while in the pre-confluent stage. The cells are then placed in solution and aerosolized onto the burn wound using a standard syringe with spray-nozzle attachment. Wood dubbed this process CellSpray (later referred to as Spray-on-Skin™) and she went on to commercialize the technology in 1993 with the help of the Australian biotechnology company Clinical Cell Culture Pty Ltd. (also referred to as C3) [102]. Over the next decade the global burn community would regard Spray-on-Skin™ with skepticism, as Wood did not publish any trials demonstrating its comparative efficacy. Wood continued to use the method routinely in her own practice and insisted that the results spoke for themselves. The lack of scientific evidence and Wood’s financial relationship with C3 would raise concerns among her peers and negatively affect global interest in the technology. Therefore, for a time, C3 proved a service, albeit on a humble scale, to burn centers in Australia, New Zealand, and the United Kingdom.

On 12th October 2002, however, Spray-on-Skin™ would return to the international limelight when militant terrorist bombed the tourist-rich Kuta district of Bali, Indonesia. The Royal Australian Air Force transported over a hundred patients to hospitals throughout Australia, of which 28 were brought to Wood’s burn unit. In preparation for such an event, Wood had coordinated with Woodside Petroleum, a local energy company, to trial a burn catastrophe response plan the year prior. As part of the plan, the production of Spray-on-Skin™ would be upregulated proportionally to meet the acute needs of the patients. By exercising this plan, Wood and her colleagues were able to save all but three of the patients they had received that day [102]. Naturally, international interest in Spray-on-Skin™ increased. In 2005, Wood introduced another technological achievement in skin transplantation technology—aerosolization of non-cultured epidermal cells. Wood called her novel system ReCell^®^; in 2008, C3 would be restructured under a new name, Avita Medical, Inc. (Valencia, California), and ReCell^®^ would be their primary product. John Harvey, the president of the Australian and New Zealand Burn Association described Wood’s work as “ground-breaking” [103]. In 2005, for her significant contributions to the field of burn care, Wood would be named Australian of the Year.

The ReCell^®^ system is available as a single-use kit that combines the extraction and application of cells into a single process. Cells are harvested from the dermal-epidermal junction of a STSG taken at 6–8/1000th of-an-inch thickness. The tissues are then enzymatically (trypsin) and mechanically disrupted. Keratinocytes, melanocytes, fibroblasts, and Langerhans cells are the suspended in a lactate solution. Each square centimeter of donor STSG generates one milliliter of suspension, which in turn can be applied to approximately 80 cm^2^ of wound area (80:1 expansion ratio) [104]. Similar to CellSpray, ReCell^®^ was introduced with little comparative evidence of efficacy relative to other routinely used skin grafting techniques. In fact, it would be Sood and colleagues who would publish the first phase 2 trial comparing outcomes between ReCell^®^ and meshed STSG in 10 patients [105]. Although under-powered, Sood showed that ReCell^®^ demonstrated similar results and required a small donor site. In 2018, a larger multi-institutional randomized control trial was performed by James Hill Holmes IV, and collaborators comparing ReCell^®^ with meshed STSG in deep partial thickness wounds in 83 patients. At four weeks, patients treated with ReCell^®^ demonstrated similar wound closure (98 vs. 100%), pain and scarring compared to controls. In contrast, control donor sites were approximately 40 times larger, incurred more pain, and expectantly took longer to heal than donor sites in the treatment group. Subsequent follow-up one year after treatment revealed that patient’s that received ReCell^®^ were considerably more satisfaction than controls [106]. Based on these trails, ReCell^®^ received approval by the Food and Drug Administration in 2018.

Critics of ReCell^®^ also raised concern over the viability of extracted cells after applications. In 2012, a study using ReCell^®^ found 75.5% of cells were viable at the time of harvest and 69.5% survived aerosolization. Critics also argued that application of non-cultured cells result in delayed epithelialization and thin wound coverage, especially when utilized on full thickness wounds that lack existing dermal elements. Initially, Wood responded to these concerns in 2007 by demonstrating the successful combined use of ReCell^®^ and Integra^®^ on a porcine model. Wood performed a single-stage repair of ten full thickness wounds on a pair of Yorkshire swine and histologically compared results with controls treated with Integra^®^ alone and ReCell^®^ alone. Wood’s results showed that simultaneous treatment enhanced reconstruction of full thickness wounds compared to controls, but once again did not provide a comparison of her technique with what would be considered a standard of care practice [107]. More than a decade later, in 2019, Holmes IV and collaborators once again performed a second multi-institutional trial comparing the concurrent use of ReCell^®^ and STSG with in-subject controls receiving STSG alone. The trial was performed on 30 patients for treatment of mixed depth wounds including deep partial thickness and full-thickness burns. The mean TBSA was 21% (+/− 13%) and the average area grafted was 2443 cm^2^ (+/− 1675 cm^2^). The results for treatment with ReCell^®^ and STSG showed non-inferiority for wound healing (92 vs. 85%) and a statistically significant reduction in donor site area (32%; *p* < 0.001) [104]. These results suggest that ReCell^®^ combined with STSG can be a safe and effective treatment for deep burn wounds and can help minimize the amount of donor surface area utilized. As of December 2020, ReCell^®^ is being used at 83 of the nations 132 burn centers and is considered one of the most promising advance in skin translocation technology [108].

## 7. The Future of Skin Grafting: The Author’s Thoughts

In the past, research has focused on optimizing expansion ratios and improving graft-related outcomes. Some would argue that the use of cultured keratinocytes and aerosolization of non-cultured skin grafts is the epitome of such pursuits. The future of burn care, therefore, will rely on the incorporation of new techniques, such as nanotechnology and 3D printing to push the boundaries of skin grafting as we know them. One of the difficulties with engineering artificial tissue is the inability to produce complex tissue layers in a reproducible manner. Computer-aided design (CAD) are used to design and produce 3D printed models. While the majority of 3D printing is done on a macro scale, CADs can be scaled down to the cellular level with the help of nanotechnology. This means it would be possible to fine-tune the composition of artificial skin substitute in a manner that is both precise and reproducible. Current techniques that are being attempted involve bioprinting cellular components onto a premade matrix and directly printing such a construct onto the burn wound.

Stefanie Michael from Hanover Medical School (Hanover, Germany) has reported promising results regarding the former technique. Utilizing Laser-assisted bioprinting (LaBP) and a mouse model, Stefanie created artificial skin using Matriderm^®^ as a foundation and printed fibroblasts and keratinocytes onto the surface. Using Ki67 as a marker for cellular growth, Michael found that his artificial skin mimics similar patterns of proliferation and differentiation found in natural skin. In the stratum basalis of normal skin, keratinocytes typically proliferate and differentiate as they progress towards the skin surface. A gradient of Ki67 was found in the artificial skin with a higher concentration at the base of the graft, mirroring the arrangement of natural skin. These grafts were only left in the mice for 11 days before being removed for examination. Apart from the proliferation patters, researchers also noted blood vessels growing towards the graft [109]. The lack of rete ridges, however, suggests the possible fragility of the artificial skin. While these findings are rudimentary in their clinical applications, they represent an early success in the possibility of 3D printed skin.

Biomedical engineer Aleksander Skardal demonstrated the latter technique of directly printing the artificial skin onto the wound. Not only did his team utilize 3D printing, but they also attempted to capitalize on the versatility and lack of immunogenicity of stem cells by incorporating them into the artificial tissue. Using a bioprinter of their own design, amniotic fluid-derived stem (AFS) cells and bone marrow-derived mesenchymal stem cells (MSC) in a fibrin-collagen gel were printed onto a wound and compared to a wound treated with a fibrin-collagen only gel in mice. The wounds were then evaluated periodically at 0, 7, and 14 days. Comparing the histology of the wounds, the researchers found greater wound closure, and re-epithelization in the stem cell treated wounds compared to the control. Histological sections revealed the AFS cell treated wound showed the greatest microvascular density and capillary diameter, while the fibrin-collagen control showed the least. Fluorescence tracking of the stem cells suggest their presence was transient and further analysis indicated that it is possible that is it the growth factors released by the stems cells which induced wound healing [110]. The direct printing of artificial skin onto a burn could be further augmented by a “portable handheld electrohydrodynamic multi-needle spray gun” described by Sofokleous making the process more accessible and convenient. Appearing similar to a hot glue gun, this device was found to be able to produce multicomponent structures with sub-micrometer precision [111]. This technology has yet to be tested printing artificial skin, but theoretically has many appealing applications. The technologies discussed here are only two of the many currently being explored in the cause of advancing the treatment of burn wounds. While many are still confined to theory and animal models, early data provide many reasons to be hopeful for their future applications in skin grafting.

In this article, we traced the history of modern skin grafting from its brutal roots in simple tissue translocation in the ancient world to modern expansion techniques with aerosolized skin cells. A timeline of this journey can be seen in Figure 5. The future of skin grafting and burn care will continue to be driven by the challenges presented by the most severely injured patients. While there is much work left to be done, reflecting on the past is liable to provoke a greater appreciation of how far modern skin grafting and burn care has already come.

## Figures and Tables

**Figure 1 medicina-57-00380-f001:**
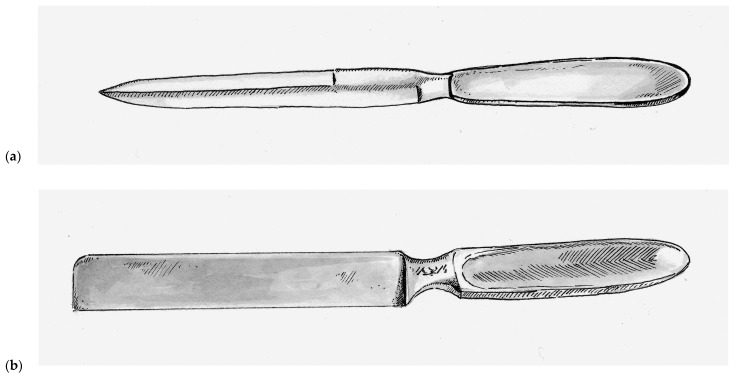
Early skin grafting knives: (**a**) The Catlin knife from the 19th century and earlier; (**b**) Thiersch’s knife in 1920.

**Figure 2 medicina-57-00380-f002:**
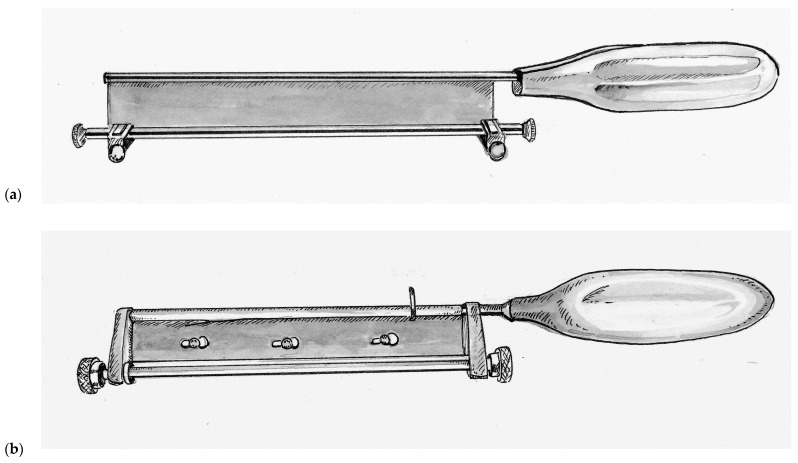
Skin grafting knives with modifiable depth: (**a**) The Humby knife in 1934; (**b**) The Watson knife in 1960.

**Figure 3 medicina-57-00380-f003:**
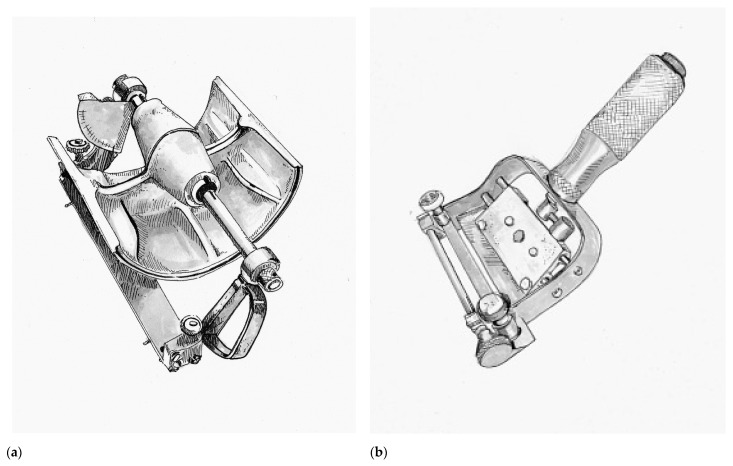
Early dermatome designs: (**a**) The Padgett–Hood dermatome in 1937; (**b**) The Brown electric dermatome in 1948.

**Figure 4 medicina-57-00380-f004:**
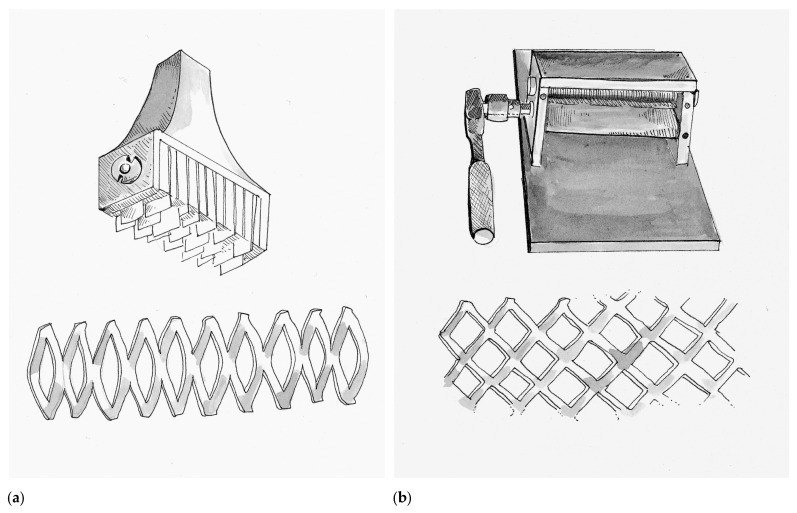
Instruments for skin graft expansion: (**a**) The Hautschlitzapparat by Lanz in 1907; (**b**) The Tanner–Vandeput mesh-dermatome in 1964.

**Figure 5 medicina-57-00380-f005:**
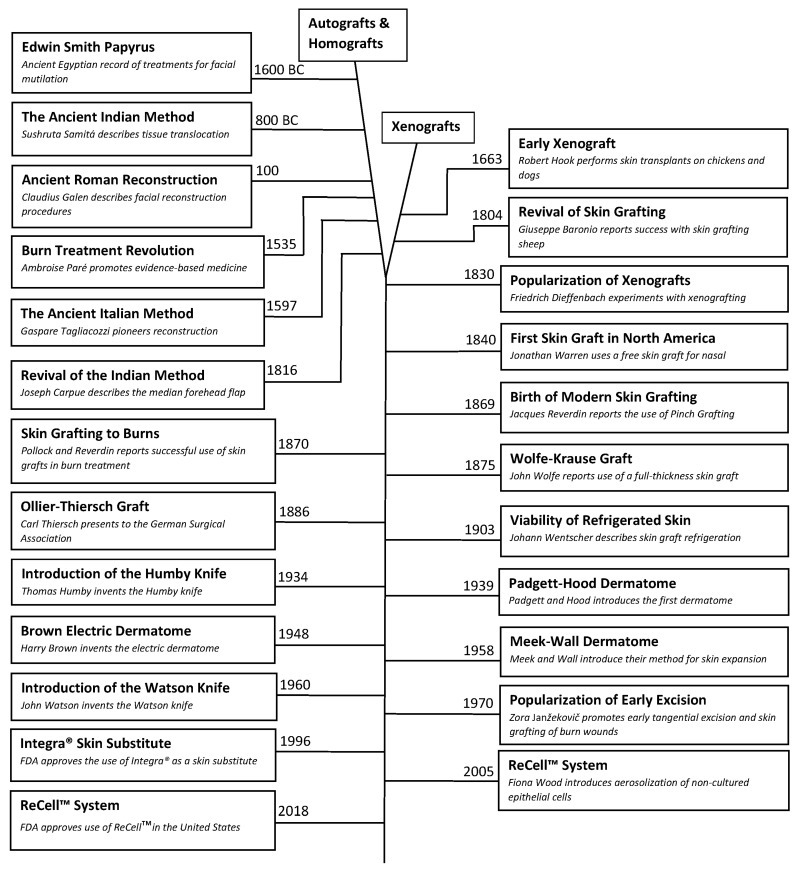
A timeline of the major historic events that preceded modern skin grafting in burn surgery.

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
