# Peer review of "A Narrative Review of the History of Skin Grafting in Burn Care"

_medicina, 2021, doi:10.3390/medicina57040380_

Round 1

Reviewer 1 Report

A very interesting paper describing the history of skin grafting in burn surgery. I would like to compliment the authors for the good work done, I really enjoyed reading this article. The only minor comment in my opinion is that the suffix": a narrative review" should be added to the title.

The article was a comprehensive narrative review describing the history of skin grafting from ancient times to nowadays. I found the article very complete and well structured. In fact, is probably one of the most interesting articles I've read in this period. I was on the fence about giving this article minor revision or accepting it in the present form. The only thing that should be changed is the fact that the review is not systematically but narrative, and this, in my opinion, should be mentioned in the title or in the text.

Thank You 

Author Response

Dear Reviewer,

Thank you for taking the time to review our submission. Thank you for your feedback and kind words. We have modified the title of the manuscript to the following in accordance with your recommendation:

"A Narrative Review of the History of Skin Grafting in Burn Surgery"

Thank you again.

Sincerely,

Deepak Ozhathil, MD

Reviewer 2 Report

This is a good read and covers the journey of skin grafting in burns well. However, despite a lot of detail in the historical journey from nose "replants" to tissue transfers to skin grafts, the review fails to cover one of the most important aspect of the history of skin grafting/tissue transplants: the immunological discoveries forming the basis of current practice.  I would recommend the authors fill this gap for completeness.

Author Response

Dear Reviewer,

Thank you for taking the time to read our manuscript and provide feedback. We have modified section 6.4 (previously Homograft) to incorporate the topic identified as missing. Section 6.4 is now titled "Homografts and Immunologic Discoveries in Skin Grafting" and discusses the important initial events in skin grafting immunology.  Three additional paragraphs and eight new references were added to the manuscript. Two previous paragraphs were merged and one reference (previously #76) was eliminated. 

Thank you again for your consideration. Please read below for the three new paragraphs.

Sincerely,

Deepak Ozhathil, MD

Around the same time across the Atlantic the Battle of Britain (1940) raged and a plane crashed in Oxford near the home of British immunologist and Zoologist, Sir Peter Medawar (1915 – 1987). The physician caring for the horribly burned pilot consulted Medawar for advice. Although Medawar had no experience caring for burn victims, he believed that skin grafting would afford the airman the best chance of survival. Unfortunately, despite Medawar’s efforts, the pilot would not survive. The experience, however, would instill in Medawar a life-long curiosity about the immunological intricacies of skin grafting. During the remainder of the war Medawar collaborated with the Scottish Plastic surgeon Thomas Gibson (1915 – 1993) to perform homografts and autografts on other soldiers [77]. The two observed that homografts, though initially appearing to incorporate, would go on to be rejected within two weeks’ time. In contrast, autografts were often successfully engrafted during the same time frame. Medawar also noticed that if a second homograft from the same donor was re-attempted, graft rejection occurred more quickly. This affirmed Medawar’s suspicions that the etiology of graft failure was immune-mediated. He reported his findings to the War Wounds Committee of the Medical Research Council in 1944, The Behaviour and Fate of Skin Autografts and Skin Homografts in Rabbits [78].

In 1945 Ray David Owens (1915 – 2014) introduced the concept of chimerism while studying dizygotic cattle twins. Owens observed the presence of “mixed blood types” that were the result of in utero genetic exposure [79]. Frank Macfarlane Burnet (1899 – 1985) an Australian Virologist built upon these findings and proposed the theory of immune tolerance, suggesting that immunologic self-awareness could be influenced, particularly during embryogenesis. Medawar tested this theory by crossing allografts between dizygotic cattle twins. He observed that grafts remained intact for several weeks longer than would typically be expected. Medawar took the experiment one step further using a mouse model. He inoculated fetal mice with splenic cells from a donor (second) mouse strain. Eight weeks later, he performed allografts using the donor strain of mice and observed that the transplanted skin was tolerated. As a control, skin from a previously unexposed (third) strain was also grafted and was expectantly rejected [80]. This experiment is credited as the foundation for modern transplant immunology and both Medawar and Burnet were awarded the Nobel Prize in Medicine for their discovery of immune tolerance

Prior to World War II, Colonel James Barrett Brown (1899 – 1971), had postulated that homograft rejection was due to the genetic disparity between donor and host. In 1937 he performed the first successful “homograft” in which both the donor and recipient of a skin graft were identical twins. During the war as the Chief of plastic surgery at Valley Forge General Hospital (Phoenixville, Pennsylvania), Brown took Joseph E. Murray (1919 – 2012), then a surgical intern and First Lieutenant under his wing. This act of charity spared Murray overseas deployment and afforded him experience caring for wartime victims of burn-related trauma. Similar to Medawar, Murray’s first-hand exposure to burn victims would inspire an academic career in tissue transplantation. Murray would lay the foundation for our modern understanding of skin’s enhanced antigenicity [81,82]. In the decades that followed World War II, a significant amount of research on the immune response to skin grafts was performed. In the 1950s, major histocompatibility complexes (MHC) were discovered. However, Rupert Everett Billingham (1921 – 2002) demonstrated that both MHC-matched and mismatched donor homografts resulted graft failure [83]. The 1960s saw a wave of animal experiments with immunosuppressants like phenothiazine derivatives, methylhydrazine derivatives, anti-lymphocyte biologics, steroids, anti-metabolites and even x-ray irradiation – none of which were effective in reducing skin graft rejection rates or were practical for use in human use [84]. In the 1970s, transplanted skin was found to generate a more robust immune response than solid organs due to its higher antigenicity. Although the exact mechanisms for cell-mediate (T-cell) and innate (NK cell) mediated acute rejection was not better understood till more recent decades, surgeons have come to understand that homograft do not replace the need for autograft, but serves as temporary bridge to autograft application. Of note, Murray replicated the work of him predecessor, Brown, performing the first living donor kidney transplant between identical twins, and was awarded the Nobel Prize in Medicine for his achievement in 1990 [81].